# Multiscale reconstruction of various vessels in the intact murine liver lobe

Qi Zhang[1], Anan Li [1,2], Siqi Chen[1], Jing Yuan [1,2], Tao Jiang[2], Xiangning Li[1,2], Qingming Luo [3], Zhao Feng [2✉] & Hui Gong [1,2✉]

The liver contains a variety of vessels and participates in miscellaneous physiological functions. While past studies generally focused on certain hepatic vessels, we simultaneously obtained all the vessels and cytoarchitectural information of the intact mouse liver lobe at single-cell resolution. Here, taking structural discrepancies of various vessels into account, we reconstruct and visualize the portal vein, hepatic vein, hepatic artery, intrahepatic bile duct, intrahepatic lymph of an intact liver lobe and peribiliary plexus in its selected local areas, providing a technology roadmap for studying the fine hepatic vascular structures and their spatial relationship, which will help research into liver diseases and evaluation of medical efficacies in the future.

[1] Britton Chance Center for Biomedical Photonics, Wuhan National Laboratory for Optoelectronics, MoE Key Laboratory for Biomedical Photonics, Huazhong University of Science and Technology, Wuhan 430074, China. [2] HUST-Suzhou Institute for Brainsmatics, Suzhou 215123, China. [3] School of Biomedical Engineering, Hainan University, Haikou 570228, China. ✉email: fengzhao@brainsmatics.org; huigong@mail.hust.edu.cn

The liver is a large organ located in the abdomen of the organism responsible for various important physiological functions, composed of plenty of lobules with similar functions and structures[1,2]. As a multifunctional accessory of the gastrointestinal tract, the liver maintains body homeostasis (especially the digestive system), including nutrient processing, protein production, energy homeostasis, and detoxification[3].

The liver is supplied by the portal vein and the hepatic artery. The hepatic artery accompanies the portal vein. The blood in the portal vein and hepatic artery enters the hepatic sinusoids, facilitating material exchange between hepatocytes and plasma, and then flows out of the liver through the hepatic vein. Besides blood vessels, bile ducts and lymphatic vessels also perform important physiological functions in the liver. The intrahepatic bile duct transports bile from hepatocytes to the intestine[4–6], surrounded by the peribiliary vascular plexus providing essential nutritious support. Intrahepatic bile ducts and lymphatic vessels also accompany the portal vein.

In a healthy liver, a relatively stable micro-ecosystem consisting of hepatocytes, extracellular matrix, and intrahepatic vessels is maintained. When the liver parenchyma suffers prolonged and repeated inflammation, fibrogenesis happens, during which large vessels such as portal vein and hepatic vein are subjected to mechanical pressure, which even induces collapse[7–10]. Also, the phenotype of liver microvessels changes from a highly specialized porous sinusoid to a continuous, more rigid capillary. Fibrosis is locally distributed around the damaged tissue, which vary among different diseases by different causes. Therefore, to study the microstructural changes during the hepatic disease process, it is necessary to obtain three-dimensional morphology of liver blood vessels, bile ducts, and lymphatic vessels in the whole liver lobe through single-cell-resolved imaging methods[11].

X-ray micro-computed tomography (μCT) combined with blood vessel casting is an important method to obtain hepatic vascular and biliary morphology. Many studies employ μCT to obtain the structures of hepatic vein, portal vein, and hepatic artery. μCT provides data support in the study of liver angiogenesis in the early stages of liver fibrosis[12], blood supply and drainage areas[13], 3D structural abnormalities of the vascular system[9], and rat blood circulation simulation[14]. However, due to the low resolution, researches based on μCT can only study the blood vessel in the liver, observation of hepatic sinusoids is limited. Phase-contrast computed tomography (PCCT) is also widely used in the study of intrahepatic vessels and bile ducts. More detailed information can be obtained by adopting PCCT imaging of local areas with higher resolution compared with μCT. PCCT is used in the study of the three-dimensional reconstruction of hepatic fibrosis samples induced by bile duct ligation and structure of proliferative bile ducts[15,16]. Due to the limitations of imaging range, PCCT has not been exploited to study hepatic sinusoids of the intact liver for the time being. The two methods mentioned above have limitations for observing the exquisite morphologies of hepatic sinusoids in an intact lobe. The current researches on hepatic sinusoids are mainly based on Synchrotron-radiation Microtomography and Deep Tissue Microscopy for high-resolution imaging of liver tissue[9,11,17]. However, restricted by imaging range, using the two methods for reconstructing and analyzing the microstructures of multi-vessels in the intact liver lobe is not reasonable.

Here, we simultaneously obtain the vascular and cytoarchitectural information of intact liver lobes with single-cell resolution, by implementing a High-definition fluorescent micro-optical sectioning tomography (HD-fMOST) system[18]. HD-fMOST combined line-illumination modulation technology with tissue sectioning, achieving a good background inhibition with high-throughput imaging. With the low background, weak signals were still discernible and rich details could be recorded, which allowed us to obtain more information from an intact liver image dataset. By cytoarchitecture, we identified hepatic vein, portal vein, and intrahepatic bile ducts. Based on the vascular information, we further identified the hepatic artery and lymphatic vessels, and reconstructed sinusoids and peribiliary plexus in selected local areas.

## Results

**Acquisition of the dataset of an intact liver lobe with single cell resolution**. We established a pipeline for hepatic vascular system acquisition procedure and obtained the liver dataset of intact liver lobes from Tek*Ai47 mice (Fig. 1a). Since the HD-fMOST imaging system implements the dual-wavelength imaging strategy, the acquired dataset consists of two image channels containing vascular information and cytoarchitecture information, respectively, with the spatial resolution of $0.32 \times 0.32 \times 1\,\mu m^3$ (Fig. 1b, c). In the red channel, namely the cytoarchitecture channel, we can observe the structures of the hepatocytes and vessel walls, implementing the co-localization of the vessel and cytoarchitecture (Fig. 1d–f). In the 100-micron projection images from the green channel, namely the vessel channel, the morphologies of vascular endothelial cells can be observed (Fig. 1g, h). We respectively used Fig. 1i, j to show the vessel channel image and the merged two channel image of the same local area of portal triads from the liver. By comparison, we can see that the intrahepatic biliary epithelial cells (BEC) are not labeled in the vessel channel (Fig. 1i). In contrast, in the cytoarchitecture channel image, we can observe the structure of hepatic cells and the morphology of BEC (Fig. 1j).

**Reconstruction of portal vein and hepatic vein in intact lobe**. We exploited the OTSU thresholding method combined with manually tuned parameters to reconstruct the portal vein and hepatic vein on the down-sampled data with a voxel size of $5 \times 5 \times 5\,\mu m^3$ (Fig. 2a). We distinguished portal vein and hepatic vein following the principle that the portal vein was accompanied by hepatic artery, bile ducts, and lymphatic vessels, while hepatic vein was not accompanied by other vessels. Based on the differences in the distribution of the portal vein and hepatic vein (Fig. 2b, c), the vessels are manually distinguished as two sets of vessels, namely the portal vein and hepatic vein, are obtained. Both portal vein and hepatic vein are similarly and parallelly distributed throughout the liver lobe with a fan-shape (Fig. 2d).

**Reconstruction of hepatic artery, bile duct, and lymphatic vessel**. Except for veins, images from two channels also provide the signals of hepatic artery, bile ducts, and lymphatic vessels which accompany the portal vein, as shown in Fig. 3a. The green channel marks the endothelial cells of blood vessels and lymphatic vessels. By comparison, we can see that the hepatic artery has a relatively thicker wall and a stable lumen than the lymphatic vessel, while lymphatic vessel wall is relatively thinner, with greater morphological variety and irregular shape, as shown in Fig. 3b. The red channel, namely the cytoarchitecture channel, marks BECs of the bile duct walls. Therefore we could observe the intrahepatic bile ducts encircled by BECs, as shown in Fig. 3c. We manually traced the hepatic artery, intrahepatic bile ducts, and lymphatic vessels. All of them accompany the portal vein, as the 3D reconstruction shown in Fig. 3d–g.

**Reconstruction of hepatic sinusoid**. In the vessel channel, we can observe the typical sinusoidal structure (Fig. 4a). We selected a data block from the liver lobe of 7-week-old mouse with the size of $200 \times 200 \times 200\,\mu m^3$. Before we reconstructed the sinusoids, we down-sampled the dataset from $0.32 \times 0.32 \times 1\,\mu m^3$ to $1 \times 1 \times 1\,\mu m^3$ for

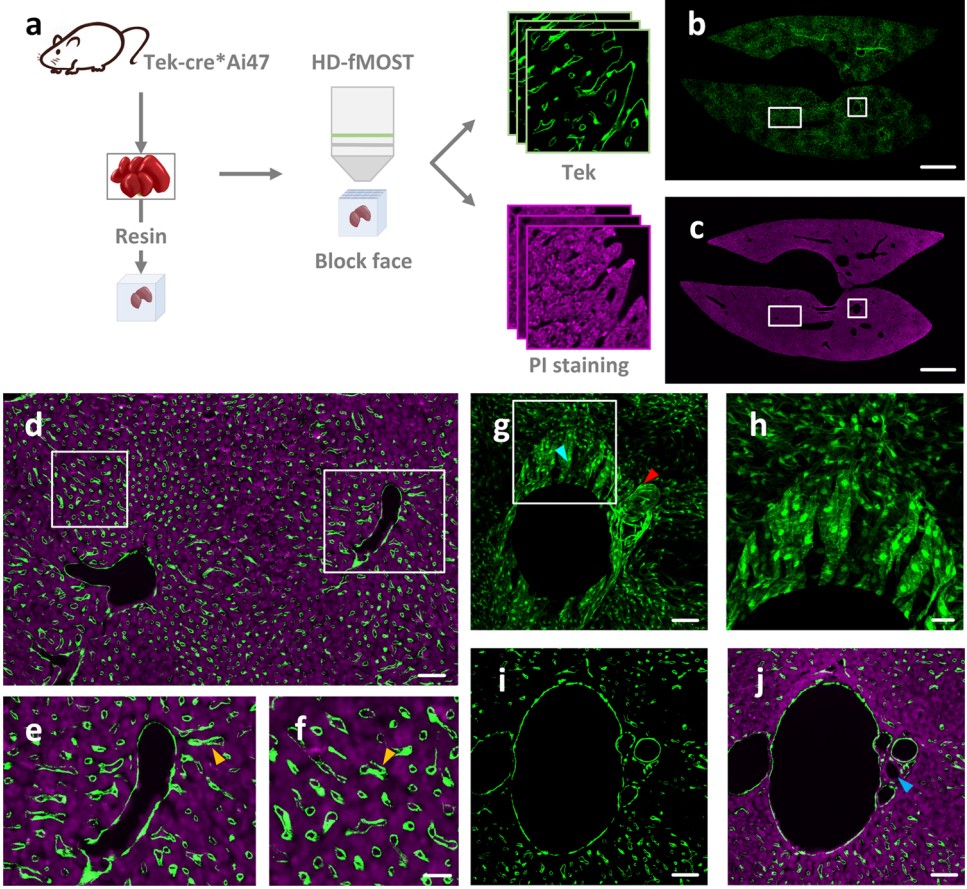

**Fig. 1 The data acquisition pipeline and acquired results. a** The procedure of acquiring dual-wavelength image dataset by high-definition fluorescent micro-optical sectioning tomography (HD-fMOST). **b** The slice image from the vascular channel of the intact mouse liver dataset. **c** The slice image from the cytoarchitecture channel of the intact mouse liver dataset. **d** The zoom-in of the merged channel images indicated in the white boxes in (**b**) and (**c**). **e**, **f** The zoom-in of the area indicated by the white boxes in (**d**). The yellow arrows indicate sinusoids. **g** The zoom-in of the merged channel images indicated in the white boxes in (**b**). The cyan arrow indicates portal vein. The red arrow indicates hepatic artery. **h** The zoom-in of the merged channel images indicated with a white box in (**g**). **i** Portal triads in vessel channel image. **j** Portal triads in merged dual-channel image. The blue arrow indicates bile ducts. The projection thickness of (**b**, **c**, **g**) is 100 μm, and the thickness of (**d**–**f**, **h**–**j**) is 1 μm. Scale bars, **b**, **c** 1 mm; **d**, **g**–**j** 50 μm; **e**, **f** 20 μm.

an isotropic spatial resolution, and then reconstructed the hepatic sinusoids within (Fig. 4b). We obtained the sinusoidal network with color-coded diameter information (Fig. 4c).

**Manually segmentation of peribiliary plexus**. In the liver dataset of the 7-week-old mouse, we selected a local data block. By merging the cytoarchitectural and vessel channel, the bile ducts and their surrounding capillary networks, named peribiliary plexus (Fig. 5a), could be observed. We used the Amira software to manually segment the structure of the peribiliary plexus (Fig. 5b, c). By observing the intersection of the peribiliary plexus with the portal vein and hepatic artery on the image, we can find that the fine structure where the blood enters the peribiliary plexus and flows into the hepatic sinusoids (Fig. 5d, e).

## Discussion

In this article, we used Tek-Cre*Ai47 mice to acquire the vessel structures and cytoarchitecture information of intact liver lobe with single-cell resolution, based on propidium iodide staining and the HD-fMOST system. According to the morphology discrepancies among different vessels, we segmented large-caliber vessels, namely portal vein, hepatic vein, hepatic artery,

intrahepatic bile ducts, and lymphatic vessels, and reconstructed hepatic sinusoids and peribiliary plexus in local areas.

There are several challenges of resolving the fine vascular structures of the intact liver. The first is how to label different vessels at a time. As a typical fluorescent nucleic acid dye, the propidium iodide is widely used in labeling the DNA and RNA, which makes it an ideal method to illuminate all cells of a tissue or even large organ in fluorescence imaging. The labeled cells include the vascular and lymphatic endothelial cells and BECs, which form remarkable textures, namely the cytoarchitecture, being able to distinguish fine tubular structures, such as portal vein, hepatic vein, bile ducts, from each other and their surroundings. Moreover, the Tek-Cre*Ai47 line is often used to mark the hepatic artery and lymphatic vessels. Putting together, we are able to simultaneously label the main kinds of hepatic vessels.

The second challenge is how to image the multiscale vessels of an intact lobe with single-cell resolution. Noting that the diameters and lengths of the hepatic vessels varies greatly, a conventional optical imaging technique may not be able to image vessels from large arteries and veins to subtle structures such as hepatic sinusoid and peribiliary plexus. However, by implementing the HD-fMOST imaging system, this barrier is surmounted by resolving the whole liver with the spatial resolution

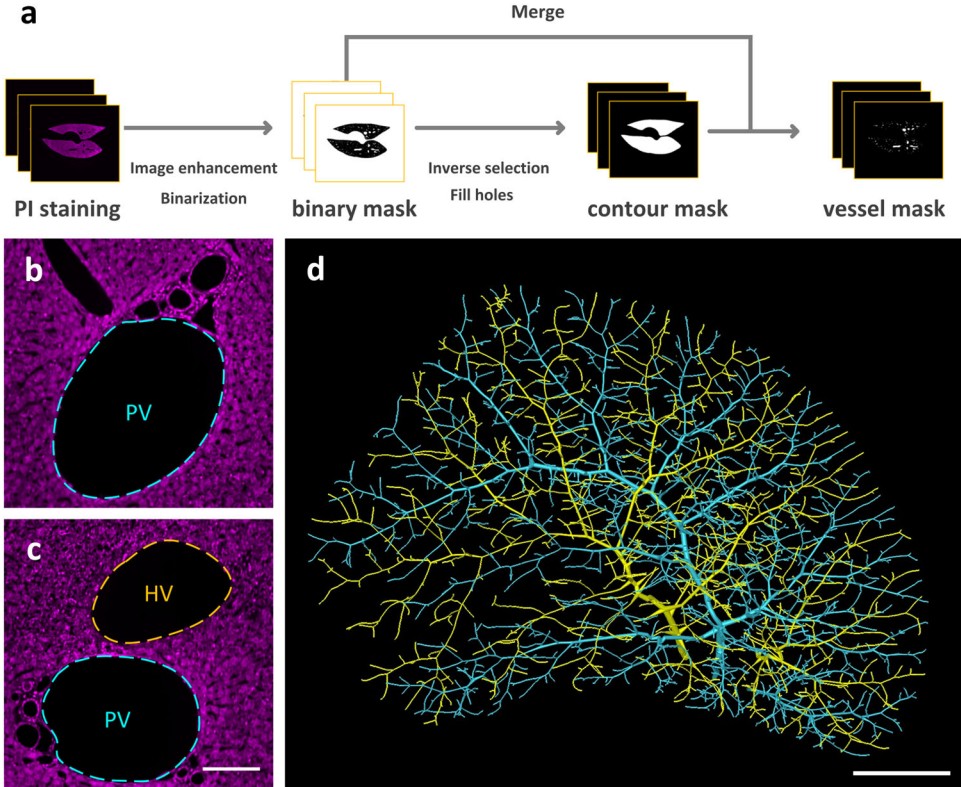

**Fig. 2 Reconstruction of the portal vein and hepatic vein. a** The reconstruction pipeline. **b**, **c** The cytoarchitecture channel images of the local areas in the liver. The portal vein and hepatic vein are respectively marked with cyan and yellow dashed lines. **d** 3D reconstruction of the portal vein and hepatic vein. The portal vein is marked with cyan. The hepatic vein is marked with yellow. PV portal vein, HV hepatic vein. Scale bar, **b**, **c** 50 μm; **d** 1 mm.

of 1 μm in both horizontal and axial directions, which is sharp enough to distinguish individual cells from each other.

In comparison, traditional researches mainly rely on μCT or PCCT imaging techniques, and acquire macro-scale dataset of intact liver or a local area dataset of the liver with micro-scale resolution, thus making it impossible to simultaneously acquire the blood vascular structures both at the intact organ level and with single-cell resolution.

Moreover, we quantitatively measured the diameter of hepatic sinusoids throughout the intact mouse liver, by counting the blood vessels with diameter less than 15 μm. This threshold is from a review of microcirculation, in which the blood vessels larger than 15 μm are treated as terminal arterioles and terminal portal venules[19]. The calculated average diameter of hepatic sinusoids in an 7-week-old mouse was $9.66 \pm 1.33\,\mu m$ ($n = 8$ biologically randomly selected local blocks), which was in accordance with the range 7–15 μm given in[19]. Also in literature[17], the diamters of periportal and pericentral hepatic sinusoids of a 6-week-old mouse, which are calculated on the Synchrotron-radiation Microtomographic images, are $8.8 \pm 2.4$ and $13.7 \pm 1.4\,\mu m$ respectively. It can be seen that our result is within the range of the previous studies.

Due to the vacuumizing during tissue embedding process, the liver tissue used in this study might expand a little, and the lumens of blood vessels would therefore be narrower compared with in-vivo status. We carefully checked the obtained images, and found that the continuity of hepatic vessels in the processed liver tissue was kept and no tissue damage was observed. Hence we are able to say that the slight shrinkage of in-vitro vessels would not affect the recognition and reconstruction of various hepatic vessels in our study.

The hepatic vascular structure is considered a key reference in studying chronic liver diseases. In chronic liver diseases, the destruction of blood vessel structure in the liver and abnormal vascular proliferation are the key points from fibrosis to cirrhosis[20]. Comprehensively understanding the abnormality and development of vascular microstructures in the process of chronic liver disease requires obtaining microvascular morphology and structure of the intact liver lobe. The technology roadmap proposed in this article enables us to observe and analyze multiscale and various types of vessels in an intact liver lobe simultaneously, which provide data basis for the judgment of chronic liver disease stages in the future.

Currently, the pipeline proposed in this article only involved the reconstruction of the macroscopic and microscopic vessels in the liver lobe though, it can be further employed to analyze the orientation and branches of the vessels as well. With the support of more liver tissue samples and the introduction of more labeling methods, we believe that the proposed pipeline will gain potential in the study of liver disease mechanisms and other research fields.

## Methods

**Animal and sample preparation**. An 7-week-old and a 25-week-old Tek-Cre*Ai47 transgenic mice were used in this research. The mice were placed in a normal cage-free of specific pathogens. The light/dark cycle was 12 h, where the temperature and the humidity were stable at 22–26 °C and 40–70%, respectively. The mice had free access to food and water. The mice were anesthetized with 1% sodium pentobarbital solution, subsequently, 0.01 M phosphate-buffered saline (Sigma-Aldrich Inc., St Louis, MO, USA) were perfused into its hearts, followed by 4% paraformaldehyde (Sigma-Aldrich) and 2.5% sucrose in 0.01 M phosphate-buffered saline. The liver lobes were dissected and fixed in 4% paraformaldehyde at 4 °C for 24 h, and then rinsed three times in a 0.01 M phosphate-buffered saline at 4 °C. It took 6 h for first two washes and 12 h for the third wash. The liver tissues were dehydrated by a graded series of ethanol solutions (50, 70, and 95%), and 100% for further dehydration for 2 h each at 4 °C. Subsequently, the tissues were sequentially immersed with Lowicryl HM20 Resin Kits (Electron Microscopy Sciences, cat.no. 14340, 50%, 70%, and 100%), and 100% embedding medium in ethanol[21]. It took 2 h for each incubation by HM20 Resin and 72 h for 100% embedding medium at 4 °C. Finally, each liver tissue was embedded in a gelatin

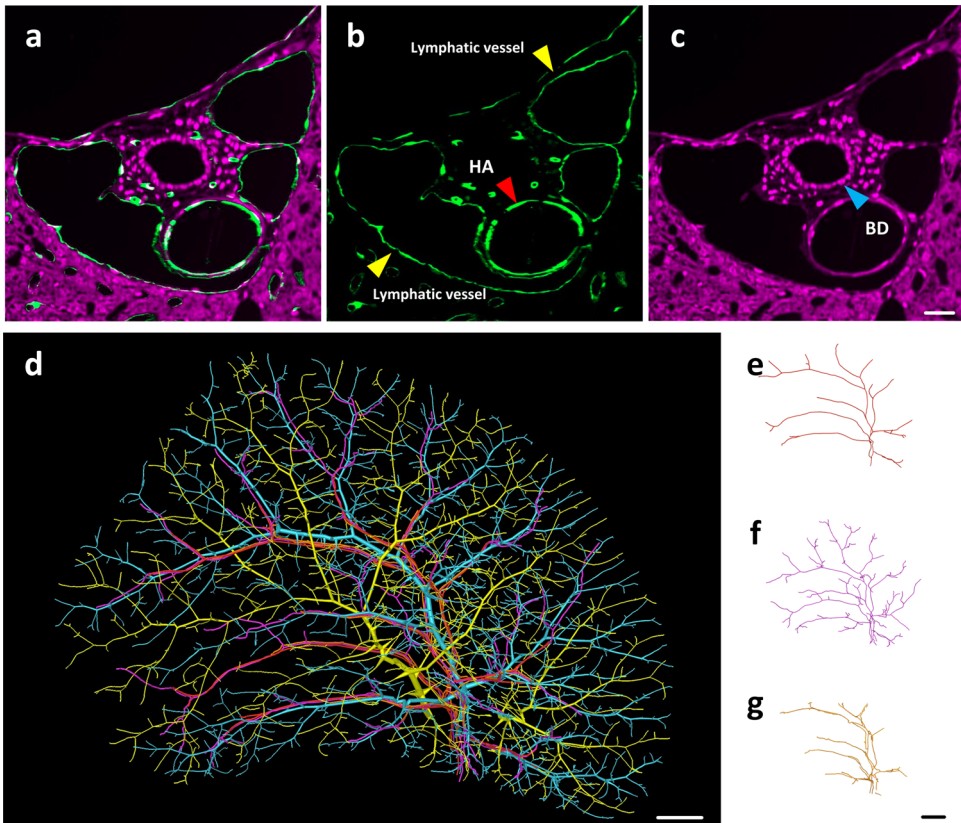

**Fig. 3 The reconstruction of the hepatic artery, bile duct, and lymphatic vessel. a** Portal triads in merged channel images. **b** Portal triads in vessel channel images. The red arrow indicates the hepatic artery. The yellow arrows indicate the lymphatic vessels. **c** Portal triads in cytoarchitecture channel images. The blue arrow indicates the bile ducts. **d** The combination of the reconstructed blood vessels, bile ducts, and lymphatic vessels in the liver. Portal vein is marked with cyan. Hepatic vein is marked with yellow. Bile ducts is marked with purple. Hepatic artery is marked with red. Intrahepatic lymphatic vessel is marked with orange. **e–g** Structure of hepatic artery, intrahepatic bile ducts, and intrahepatic lymphatic vessel are shown separately. HA hepatic artery, BD bile duct. Scale bar, **a–c** 20 μm, **d–g** 1 mm.

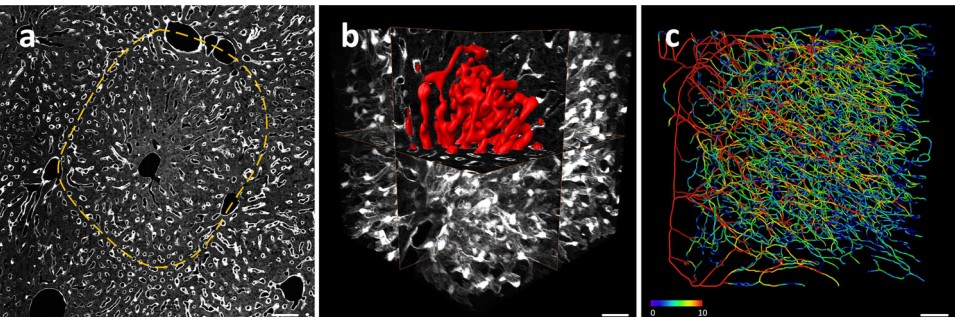

**Fig. 4 Quantitative analysis of hepatic sinusoids. a** A typical hepatic sinusoidal structure, marked by yellow dash-line. **b** The reconstruction of hepatic sinusoidal in the local area. **c** 3D visualization of sinusoid network, the diameter is encoded by different colors. Scale bar, 50 μm (**a**), 20 μm (**b**, **c**).Color bar, unit: μm (**c**).

capsule, which was filled with HM20 and polymerized at 50 °C for 24 h. All animal experiments followed procedures approved by the Institutional Animal Ethics Committee of Huazhong University of Science and Technology, and all experiments were carried out in accordance with relevant guidelines and regulations.

**Imaging**. The HD-fMOST system was used to perform dual-wavelength imaging of the Tek-cre*Ai47 liver lobe. The green Channel acquired the fluorescent signals of vascular endothelial cells, and the red channel acqcuired propidium iodide labeled cytoarchitecture information. With HD-fMOST, we simultaneously obtain vessel structures and cytoarchitecture information in an intact liver lobe at single-cell resolution. HD-fMOST is an imaging technology, which has been successfully used in the study of neuronal circuits in the brain[18]. The dataset acquired from the 25-

week-old mouse contains 6236 coronal planes with the data size of $11115 \times 9600 \times 6236 \ \mu m^3$, while the dataset from the 7-week-old mouse contains 10312 coronal planes with the data size of $9786 \times 8640 \times 10312 \ \mu m^3$. The uncompressed data volume of the two acquired datasets is 17.7 TB and 23.2 TB respectively. Acquiring the image dataset of the 7-week-old mouse costs 6 days while for the 25-week-old mouse 4 days and 20 h.

**Image preprocessing**. To reconstruct the large-caliber vessels in the liver, median filter was performed on the original resolution datasets for denoising, followed by gamma correction to enhance image contrast. The acquired two channel images were both bicubically down-sampled to $5 \times 5 \times 5 \ \mu m^3$. To trace and correct thinner blood vessels, lymphatic vessels, and bile ducts, we converted the original-

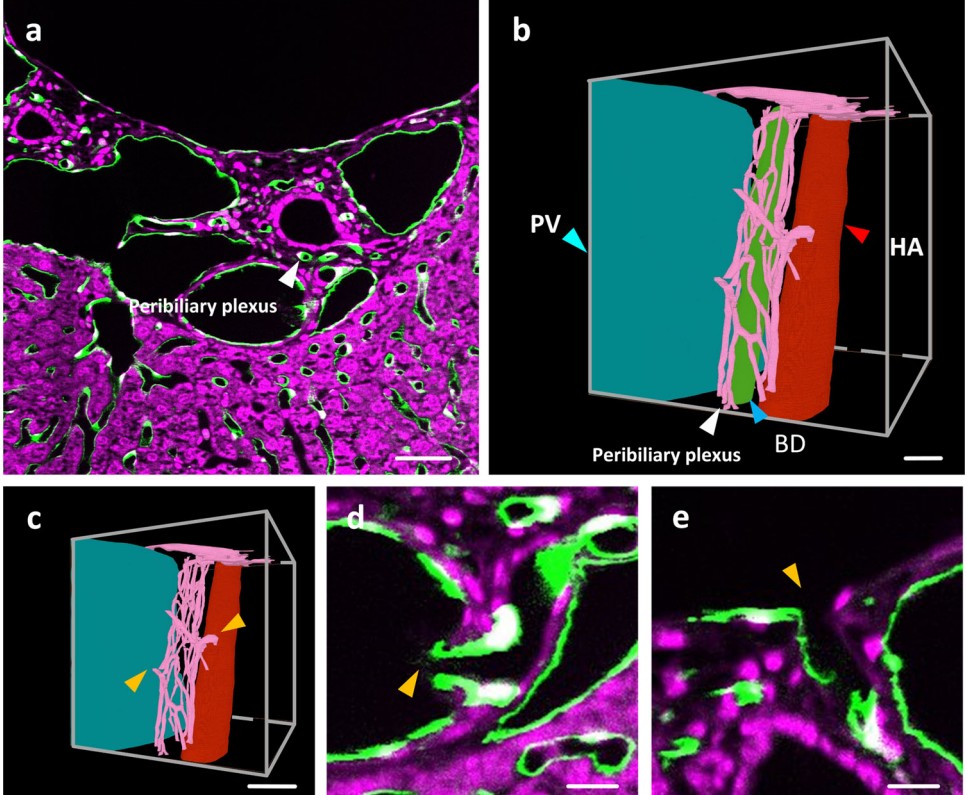

**Fig. 5 The reconstruction of the peribiliary plexus. a** The vessel distribution of the portal triads. Bile ducts and peribiliary are pointed with the white arrow. **b** Reconstruction of vessels in a selected local block. Portal vein, hepatic artery, bile ducts, and peribiliary plexus are respectively marked in blue, red, green, and pink. The cyan arrow indicates portal vein. The red arrow indicates hepatic artery. The white arrow indicates peribiliary plexus. The blue arrow indicates bile ducts. **c** The intersection of the peribiliary vascular plexus with the hepatic artery and portal vein is marked using yellow arrows. **d** The merged channel image at the intersection of hepatic artery and peribiliary vascular plexus indicated with the upper right arrow in (**c**). The intersection is marked with the yellow arrows. **e** The local area of merged channel images at the intersection between portal vein and peribiliary vascular plexus indicated with the lower left arrow in (**c**). The intersection is marked with yellow arrow. PV portal vein, HA hepatic vein, BD bile duct. Scale bar, 25 μm (**a, b**), 50 μm (**c**), 10 μm (**d, e**).

resolution dataset into multi-resolution organized TDat format[22]. We extracted several data blocks from the TDat-formatted dataset with the resolution of 1.28 × 1.28 × 4 μm³ and the size of 2000 × 2000 × 2000 μm³. The spatial distribution of the extracted blocks covered the range of the entire liver lobe, which was used for further visualization and analysis.

**Vessel reconstruction**. On the cytoarchitecture images from the red channel, we combined the OTSU thresholding method with manual-corrected parameters to reconstruct the hepatic vein and portal vein of the mouse liver (Fig. 2a). NeuronStudio[23] (v 0.9.92) software was then used to skeletonize the vessels within. Afterward, Amira software (v.6.1.1, FEI) was used to distinguish hepatic vein and portal vein, and to correct errors such as missing or misidentified branches based on the cytoarchitecture information. Manual tracing of hepatic artery, bile ducts, and lymphatic vessels was also executed with the Amira. On the vessel images provided by the green channel, we selected several data blocks with the size of 200 × 200 × 200 μm³. We reconstructed the hepatic sinusoids using the parameter-corrected Otsu thresholding method combined with morphological operations. The skeletonization of sinusoids was performed with the Amira software. The detection of the diameter of hepatic sinusoids was executed with the rayburst algorithm[24]. We used Amira to manually reconstruct the local peribiliary vascular plexus on the merged images of two channels. The filament editor module, segmentation editor, and autoskeleton tool of Amira software were respectively used in vessel tracing, manual segmentation of peribiliary plexus, and sinusoidal skeletonization. The time cost for reconstructing portal vein and hepatic vein is about 3 days. Manual tracing of hepatic artery, bile ducts, and lymphatic vessels takes around 2 weeks.

**Statistics and reproducibility**. To calculate the mean diameter of hepatic sinusoids from a 7-week mice, we randomly selected 8 data blocks from one dataset without any preference. The result was shown as mean value ± standard deviation.

**Reporting summary**. Further information on research design is available in the Nature Research Reporting Summary linked to this article.

## Data availability

The data generated during the current study are available from the corresponding author on reasonable request.

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

## Acknowledgements

We appreciate the MOST group members of the Britton Chance Center for Biomedical Photonics for assistance with literature collection and comments on the manuscript. This work was financially supported by the National Key Research and Development Program of China (grant No. 2017YFA0700402), National Natural Science Fundation of China (grant No. 61890953), CAMS Innovation Fund for Medical Sciences 2019-I2M-5-014, and Suzhou Science and Technology Development Program (grant No. SYG201915).

## Author contributions

Q.Z. segmented the hepatic vessels, analyzed the data, and wrote the manuscript. A.L. revised the content of the manuscript. S.C. and X.L. prepared the mouse liver sample. T.J. and J.Y. acquired the image dataset. Q.L. revised the figures and content of the manuscript. Z.F. modified the content, structure, and spellings of the manuscript. H.G. conceived and designed the article and coordinated activities from all authors.

## Competing interests

The authors declare no competing interests.
