## [Peer Review File · Communications Biology]

Reviewers' comments:

Reviewer #1 (Remarks to the Author):

It is of great interest to study the simultaneous acquisition of all vascular and cellular structural information of intact mouse cytoarchitectural information at single-cell resolution. In particular, I believe that reconstructing and visualizing microscopic vestibules in the liver lobe will provide a technological roadmap for the study, which will be helpful in the study of liver disease. I think the paper is sound and should be published, however I have some questions that really need to be addressed before the paper can be considered for publication:

1. Line 1. What does "multisacle" in the title mean? Multiscale is not well heard in the paper. (Do you mean multiscale by using several types for OBJ or did you use multiscale in the sense of using two channels?)
2. Line 41-46. You mentioned that CT had a low resolution, making it difficult to see hepatic sinusoids. In 2013, in the following paper, hepatic sinusoid imaging was performed with uCT. If there is another advantage compared to uCT, it would be good if you describe it.
Yoon YJ, Chang S, Kim OY, et al. Three-dimensional imaging of hepatic sinusoids in mice using synchrotron radiation micro-computed tomography. PLoS One. 2013;8(7):e68600. Published 2013 Jul 5. doi:10.1371/journal.pone.0068600
3. Line 50-51, 55 "Due to the limitations of imaging range" It is difficult to understand what "the limitation of imaging range" means by looking at the preceding sentence. Detailed explanation please.
4. Line 59-60. There seems to be a lack of High-definition Fluorescent Micro-Optical Sectioning Tomography(HD-fMOST) description and performance description. Although a reference is attached, a brief explanation would be better for readers to understand.
5. Figure 2. Any information on the actual sample size? How did you combine each slice image and how much overlap was it? Also, is the image on one side subjected to mosaic processing? What is the imaging FOV?
6. Line 97. Figure 2.(C) Yellow dashed lines is Hepatic vein. It seems to have been marked incorrectly. (PV -> HV)
7. Line 123. Why is the voxel size in 25-week-old transgenic mice(Fig. 1-3) different from the voxel size in 7-week old mice(Fig. 4-5)?
8. Line 202. I think the paper needs information about what kind of embeddings were made.
9. line 216-218. Are the data sizes of 7-week-old and 25-week-old mice the same? If the week is different, the size of the liver is also different, so I think the data size will also be different. Doesn't the information indicated in the paper refer to one of the above two data? If so, it seems that it will be necessary to enter the information of the other week-mice data
10. line 227 You mentioned that " the size of 2000*2000 *2000 μm^3 ". "Each channel contains 6236 coronal planes, with a voxel size of 0.32 *0.32 *1 μm^3 " If we calculate based on the above, it should come out like this "the size of 2000 *2000 *6200 μm^3 " what's your opinion? Is there another process?

Reviewer #2 (Remarks to the Author):

This manuscript reports a novel imaging method to simultaneously obtain all the vessels and cytoarchitectural information of mouse liver lobe at single-cell resolution. This method enables reconstruction and visualization of the 3D structures of portal vein, hepatic vein, hepatic artery, intrahepatic bile duct, intrahepatic lymph and peribiliary plexus. Therefore it will be useful to study the fine hepatic vascular structures and their spatial relationship, which will help to research the

liver biology and pathology. This paper can be accepted after addressing the following minor comments.

1. Fig. 2c, both vessels are labeled with "PV". Should one of them be "HV"?
2. How to distinguish PV from HV?
3. It will be helpful to report the typical amount of time to obtain 3D images (imaging), as well as segmentation and reconstruction of 3D volumes.

For Referee #1:

It is of great interest to study the simultaneous acquisition of all vascular and cellular structural information of intact mouse cytoarchitectural information at single-cell resolution. In particular, I believe that reconstructing and visualizing microscopic vestibules in the liver lobe will provide a technological roadmap for the study, which will be helpful in the study of liver disease. I think the paper is sound and should be published, however I have some questions that really need to be addressed before the paper can be consider for publication:

Comment #1: Line 1. What does "multiscale" in the title mean? Multiscale is not well heard in the paper.(Do you mean multiscale by using several types for OBJ or did you use multiscale in the sense of using two channels?	Response #1: We use "multiscale" to indicate that we acquired both macro- and micro- scale vessels in the liver. The macroscale vessels reconstructed in this paper include hepatic vein, portal vein, hepatic artery, bile ducts and lymphatic vessels, and the microscale vessels include hepatic sinusoids and peribiliary plexus.
Comment #2: Line 41-46. You mentioned that CT had a low resolution, making it difficult to see hepatic sinusoids. In 2013, in the following paper, hepatic sinusoid imaging was performed with uCT. If there is another advantage compared to uCT, it would be good if you describe it. Yoon YJ, Chang S, Kim OY, et al. Three-dimensional imaging of hepatic sinusoids in mice using synchrotron radiation micro-computed tomography. PLoS One. 2013;8(7):e68600. Published 2013 Jul 5. doi:10.1371/journal.pone.0068600	Response #2: The synchrotron radiation micro-computed tomography (SNRμCT) is considered as an important comparison with our study in the manuscript. This technology performs a high-definition imaging capable of observing subtle structures such as hepatic sinusoids. The limitation of SNRμCT is that it could only image the sinusoids of a local area, while the proposed method in this manuscript could obtain the hepatic sinusoids of an intact liver lobe at one time. As a supplementation, we cited the mentioned paper in Line 55.
Comment #3: Line 50-51, 55 "Due to the limitations of imaging range" It is difficult to understand what" the limitation of imaging range" means by looking at the preceding sentence. Detailed explanation please.	Response #3: Thanks for your comment. What we want to express here is that constraint by its imaging principle, there is a compromise between the imaging range and spatial resolution of PCCT. If PCCT is used to acquire the fine structures such as hepatic sinusoids with high spatial resolution, the imaging range will be limited to a local area, preventing it from imaging the intact liver at one time.
Comment #4: Line 59-60. There seems to be a lack of	Response #4: HD-fMOST combined line-illumination modulation (LiMo) technology with tissue sectioning,

High-definition Fluorescent Micro-Optical Sectioning Tomography(HD-fMOST) description and performance description. Although a reference is attached, a brief explanation would be better for readers to understand.	achieving a good background inhibition with high-throughput imaging. With the low background, weak signals were still discernible and rich details could be recorded, which allowed us to obtain more information from an intact liver image dataset. We added this short description of HD-fMOST in Line 59-62.
Comment #5: Figure 2. Any information on the actual sample size? How did you combine each slice image and how much overlap was it? Also, is the image on one side subjected to mosaic processing? What is the imaging FOV?	Response #5: The original images collected from HD-fMOST were 2048-pixel-width strips with 1-μm-thick. By stitching the strips and removing the neighboring overlaps, the coronal images were created. The overlap was 5-pixel-width. The image on one side is subjected to mosaic processing, and the imaging FOV is 655.36 μm width.
Comment #6: Line 97. Figure 2.(C) Yellow dashed lines is Hepatic vein. It seems to have been marked incorrectly. (PV -> HV)	Response #6: Thanks, we have already corrected the mark as hepatic vein in Fig. 2c. Comment #7: Line 123. Why is the voxel size in 25-week-old transgenic mice(Fig. 1-3) different from the voxel size in 7-week old mice(Fig. 4-5)?	Response #7: Thank you for the reminding. The two mice with different ages used in our experiment were both imaged with the voxel size of $0.32 \times 0.32 \times 1 \mu\text{m}^3$. However, before we reconstructed the sinusoids of 7-week-old mouse, we down-sampled the dataset from $0.32 \times 0.32 \times 1 \mu\text{m}^3$ to $1 \times 1 \times 1 \mu\text{m}^3$ for an isotropic spatial resolution, which would benefit the reconstruction. The lack of necessary description in the manuscript may lead to this misunderstanding, so we supplemented a description of the down-sampling operation in Line 126-128.

Comment #8: Line 202. I think the paper needs information about what kind of embeddings were made.	Response #8: We embedded the liver tissues with Lowicryl HM20. The detailed procedure is as follows. The liver tissue were rinsed 3 times in a 0.01 M PBS solution (Sigma-Aldrich) at 4 °C. It took 6 hours for first two washes and 12 hours for the third wash. The liver tissues were dehydrated by a graded series of ethanol solutions (50%, 70%, 95%), and 100% for further dehydration for 2 hours each at 4 °C. Subsequently, the tissues were sequentially immersed with Lowicryl HM20 Resin Kits (Electron Microscopy Sciences, cat.no. 14340, 50%, 70%, 100%), and 100% embedding medium in ethanol. It took 2 hours for each incubation by HM20 Resin and 72 hours for 100% embedding medium at 4 °C. Finally, each liver tissue was embedded in a gelatin capsule, which was filled with HM20 and polymerized at 50 °C for 24 h. We replaced “and rinsed in PBS at 4°C for 12 h for following imaging procedures” with the detailed description above and cited the article of HM20 embedding in Line 214-221.
Comment #9: line 216-218. Are the data sizes of 7-week-old and 25-week-old mice the same? If the week is different, the size of the liver is also different, so I think the data size will also be different. Doesn't the information indicated in the paper refer to one of the above two data? If so, it seems that it will be necessary to enter the information of the other week-mice data	Response #9: Yes, the data sizes of two datasets are different. The dataset acquired from the 25-week-old mouse contains 6236 coronal planes with the data size of $11115 \times 9600 \times 6236 \mu\text{m}^3$, while the dataset from the 7-week-old mouse contains 10312 coronal planes with the data size of $9786 \times 8640 \times 10312 \mu\text{m}^3$. Noticing that the large discrepancy between the numbers of coronal planes is the result of different orientation of the samples. We have added the description of different data sizes in Line 231-233.
Comment #10: line 227 You mentioned that ” the size of $2000 \times 2000 \times 2000 \mu\text{m}^3$ “. “Each channel contains 6236 coronal planes, with a voxel size of $0.32 \times 0.32 \times 1 \mu\text{m}^3$” If we calculate based on the above, it should come out like this “the size of $2000 \times 2000 \times 6200 \mu\text{m}^3$ “ what's your opinion? Is there another process?	Response #10: Thanks for your comment. The acquired dataset contains 6236 coronal planes, the size of each plane is 34736×30000 pixels. Multiplying the voxel size of $0.32 \times 0.32 \times 1 \mu\text{m}^3$, the total size of the dataset is $11115 \times 9600 \times 6236 \mu\text{m}^3$. However, this data volume reaches 9.19 TB, which is too large to fully load into the Amira software used for manual tracing throughout the intact liver lobe. Therefore, as written in the manuscript, we selected from the intact dataset a series of local data blocks with the size of $2000 \times 2000 \times 2000 \mu\text{m}^3$. We

	choose this size since a larger one might cause out of memory problem. After we finished the tracing procedure of every data block, we combine the tracing results together. As we reserved overlaps among the selected local data blocks, the results could be combined satisfactorily.
--	---

For Referee #2:

This manuscript reports a novel imaging method to simultaneously obtain all the vessels and cytoarchitectural information of mouse liver lobe at single-cell resolution. This method enables reconstruction and visualization of the 3D structures of portal vein, hepatic vein, hepatic artery, intrahepatic bile duct, intrahepatic lymph and peribiliary plexus. Therefore it will be useful to study the fine hepatic vascular structures and their spatial relationship, which will help to research the liver biology and pathology. This paper can be accepted after addressing the following minor comments.

Comment #1: Fig. 2c, both vessels are labeled with “PV”. Should one of them be “HV”?	Response #1: Thanks, we have already corrected the mark as hepatic vein in Fig. 2c. Comment #2: How to distinguish PV from HV?	Response #2: We distinguished PV and HV following the principle that the PV was accompanied by HA, BD and lymphatic vessels, while HV was not accompanied by other vessels. We added the description of how we distinguished PV and HV in Line 94-96.
Comment #3: It will be helpful to report the typical amount of time to obtain 3D images (imaging), as well as segmentation and reconstruction of 3D volumes.	Response #3: Acquiring the image dataset of the 7-week-old mouse costs 6 days while for the 25-week-old mouse 4 days and 20 hours. The time cost for reconstructing PV and HV is about 3 days. Manual tracing of HA, BD and lymphatic vessels takes around 2 weeks. The time cost of these procedures was supplemented in Line 233-234, 257-259, respectively.